# ATLAS: Universal Function Approximator for Memory Retention

## Abstract

Artificial neural networks (ANNs), despite their universal function approximation
capability and practical success, are subject to catastrophic forgetting. Catastrophic
forgetting refers to the abrupt unlearning of a previous task when a new task is
learned. It is an emergent phenomenon that plagues ANNs and hinders continual
learning. Existing universal function approximation theorems for ANNs guarantee
function approximation ability, but seldom touch on the model details and do not
predict catastrophic forgetting. This paper presents a novel universal approximation
theorem for multi-variable functions using only single-variable functions and
exponential functions. Furthermore, we present *ATLAS*—a novel ANN architecture
based on the exponential approximation theorem and B-splines. It is shown that
ATLAS is a universal function approximator capable of memory retention and,
therefore, continual learning. The memory retention of ATLAS is imperfect,
with some off-target effects during continual learning, but it is well-behaved and
predictable. An efficient implementation of ATLAS is provided. Experiments
are conducted to evaluate both the function approximation and memory retention
capabilities of ATLAS.

## 1  Introduction

Catastrophic forgetting [7, 13, 23] is an emergent phenomenon where a machine learning model
such as an artificial neural network (ANN) learns a new task, and the subsequent parameter updates
interfere with the model's performance on previously learned tasks. Catastrophic forgetting is also
called catastrophic interference [19]. If an ANN cannot effectively learn many tasks, it has limited
utility in the context of continual learning [9, 12]. Catastrophic forgetting is like learning to pick
up a cup, but simultaneously forgetting how to breathe. Even linear functions are susceptible to
catastrophic forgetting, as illustrated in Figure 1

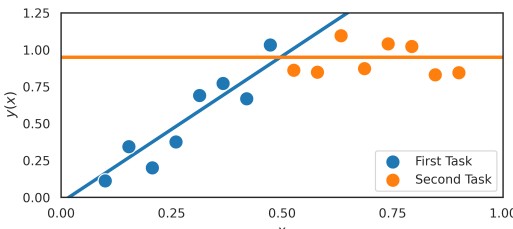

Figure 1: A linear function is susceptible to catastrophic forgetting.

The simple example of a linear regression model being susceptible to catastrophic forgetting might be due to the non-linearity of the target function, noise, or parameter sharing across the input. Parameter sharing is avoidable with piece-wise defined functions such as splines [27]. ANNs can be explained in many ways; a useful analogy is to compare ANNs to very large lookup tables that store information. Removing and updating values has off-target effects throughout the table or ANN.

Universal function approximation theorems are a cornerstone of machine learning, and prove that ANNs can approximate any given continuous target function [10, 11, 15] under certain assumptions. The theorems do not specify how to find an ANN with sufficient performance for problems in practice. Gradient descent optimisation is the convention for finding/training neural networks, but other optimisation and learning procedures exist [22]. ATLAS models trained with gradient descent methods exhibit desirable properties. However, other optimisation techniques like evolutionary algorithms may not elicit the same properties.

This paper introduces ATLAS—a novel universal function approximator based on B-splines that has some intrinsic memory retention, even in the absence of other training and regularisation techniques. ATLAS has well-behaved parameter gradients that are sparse, bounded and orthogonal between input points that are far enough from each other. The accompanying representation and universal approximation theorems are also provided.

## 2 Relevant Studies

It is conjectured that overlapping representations in ANNs lead to catastrophic forgetting [12]. Catastrophic forgetting occurs when parameters necessary for one task change while training to meet the objectives of another task [14, 20]. The least desirable strategy to mitigate catastrophic forgetting is retraining a model over all tasks. Regularisation techniques like elastic weight consolidation (EWC) have also been employed [14]. Data augmentation approaches such as rehearsal and pseudo-rehearsal have also been employed [23]. Other ideas from optimal control theory in combination with dynamic programming have also been applied to counteract catastrophic forgetting, with a cost functional similar in form to the action integral from physics and Lagrangian mechanics [16].

Orthogonal Gradient Descent (OGD) is a training augmentation or optimisation technique that modifies the gradient updates of subsequent tasks to be orthogonal to previous tasks [6, 1]. One can describe data in terms of a distribution defined over the input space, target values, and time (the order of data or tasks that are presented during training). OGD attempts to make gradient updates orthogonal to each other over time. ATLAS, in contrast, possesses distal orthogonality, meaning that if two inputs are far enough from each other in the input space, then corresponding gradient updates will be orthogonal. A corollary of this is that if the data distribution between tasks shifts in the input space, then the subsequent gradient updates will tend to be orthogonal. ATLAS does not use external memory like OGD. Extensions of OGD include PCA-OGD, which compresses gradient updates into principal components to reduce memory requirements [4]. The Neural Tangent Kernel (NTK) overlap matrices, as discussed by Doan et al. [4], could be a useful tool for analysing ATLAS models.

The survey by Delange et al. [3] gives an extensive overview of continual learning to address catastrophic forgetting. ATLAS is a model that implements parameter isolation, because of its use of piece-wise defined splines. Particularly relevant to ATLAS is the work on scale of initialisation and extreme memorisation [21]. Increasing the density of basis functions in ATLAS can lead to better memorisation, and increases the scale of some parameters in ATLAS which may affect generalisation.

Pi-sigma neural networks use nodes that compute products instead of sums [26]. Pi-sigma neural networks have some similarities with the global structure of ATLAS. B-splines, which form the basis of ATLAS, have been applied for machine learning [5]. Scardapane et al. [25] investigated trainable activation functions parameterised by splines. Uniform cubic B-splines have basis functions that are translates of one another [2]. Uniform cubic B-splines have been tested for memory retention, and ATLAS is an improvement on existing spline models [27].

B-splines, and by extension ATLAS, can be trained to fit lower frequency components, expanded and trained again until a network is found with sufficient accuracy and generalisation, similar to other techniques [17, 18]. It is not necessary to expand the capacity of an ATLAS model to learn new tasks, as with some other approaches [24]. ATLAS does in practice demonstrate something akin to "graceful forgetting" as discussed in Golkar et al. [8].

## 3   Notation

Vector quantities like $\vec{\mathbf{x}}$ are clearly indicated with a bar or arrow for legibility. Parameters, inputs, functions etc. without a bar or arrow are scalar quantities like $S(x)$. Some scalar quantities with indices are the scalar components of a vector like $x_j$ or scalar parameters in the model like $\theta_i$. The gradient operator that acts on a scalar function like $\vec{\boldsymbol{\nabla}}_{\vec{\theta}} A(\vec{\mathbf{x}})$ yields a vector-valued function $\vec{\boldsymbol{\nabla}}_{\vec{\theta}} A(\vec{\mathbf{x}})$ as is typical of multi-variable calculus.

## 4   Exponential Representation Theorem

Any continuous multi-variable function on a compact space can be uniformly approximated with multi-variable polynomials by the Stone-Weierstrass Theorem. Let $\mathcal{I}$ denote an index set of tuples of natural numbers including zero such that $i_j \in \mathbb{N}^0$ for all $j \in \mathbb{N}$ with $i = (i_1, .., i_n) \in \mathcal{I}$ and $a_i \in \mathbb{R}$. Multi-variable polynomials can be represented as:

$$y(\vec{\mathbf{x}}) = y(x_1, .., x_n) = \sum_{i \in \mathcal{I}} a_i x_1^{i_1} x_2^{i_2} ... x_n^{i_n} = \sum_{i \in \mathcal{I}} a_i \Pi_{j=1}^n x_j^{i_j}$$

Each monomial term $a_i \Pi_{j=1}^n x_j^{i_j}$ is a product of single-variable functions in each variable. It is desirable to rewrite products as sums using exponentials and logarithms.

**Lemma 1.** *For any $a_i \in \mathbb{R}$, there exists $\gamma_i > 0$ and $\beta_i > 0$, such that: $a_i = \gamma_i - \beta_i$*

**Theorem 1** (Exponential representation theorem). *Any multi-variable polynomial function $y(\vec{\mathbf{x}})$ of $n$ variables over the positive orthant, can be exactly represented by continuous single-variable functions $g_{i,j}(x_j)$ and $h_{i,j}(x_j)$ in the form:*

$$y(\vec{\mathbf{x}}) = \sum_{i \in \mathcal{I}} \exp\big(\Sigma_{j=1}^n g_{i,j}(x_j)\big) - \exp\big(\Sigma_{j=1}^n h_{i,j}(x_j)\big)$$

*Proof.* Consider any monomial term $a_i \Pi_{j=1}^n x_j^{i_j}$ with $a_i \in \mathbb{R}$, then by Lemma 1 there exist strictly positive numbers $\gamma_i > 0$ and $\beta_i > 0$, such that:

$$
\begin{aligned}
a_i \Pi_{j=1}^n x_j^{i_j} &= \gamma_i \Pi_{j=1}^n x_j^{i_j} - \beta_i \Pi_{j=1}^n x_j^{i_j} \\
&= \exp\Big(\log\Big(\gamma_i \Pi_{j=1}^n x_j^{i_j}\Big)\Big) - \exp\Big(\log\Big(\beta_i \Pi_{j=1}^n x_j^{i_j}\Big)\Big) \\
&= \exp\Big(\log(\gamma_i) + \Sigma_{j=1}^n \log\Big(x_j^{i_j}\Big)\Big) - \exp\Big(\log(\beta_i) + \Sigma_{j=1}^n \log\Big(x_j^{i_j}\Big)\Big)
\end{aligned}
$$

The argument of each exponential function is a sum of single-variable functions and constants. Without loss of generality, a set of single-variable functions can be defined such that:

$$a_i \Pi_{j=1}^n x_j^{i_j} = \exp\big(\Sigma_{j=1}^n g_{i,j}(x_j)\big) - \exp\big(\Sigma_{j=1}^n h_{i,j}(x_j)\big)$$

Since this holds for any $a_i \Pi_{j=1}^n x_j^{i_j}$ and all $i \in \mathcal{I}$, it follows that:

$$y(\vec{\mathbf{x}}) = \sum_{i \in \mathcal{I}} \exp\big(\Sigma_{j=1}^n g_{i,j}(x_j)\big) - \exp\big(\Sigma_{j=1}^n h_{i,j}(x_j)\big)$$

$\square$

This result is fundamental to the paper. Since every continuous function can be approximated with multi-variable polynomials, it follows that every continuous function can be approximated with positive and negative exponential functions. Single-variable function approximators are pivotal and must be reconsidered. Universal function approximation can also be proven with the sub-algebra formulation of the Stone-Weierstrass theorem, but it's not as delightful and simple as the first constructive proof given above.

## 5 Single-Variable Function Approximation

Splines are piece-wise defined single-variable functions over some interval. Each sub-interval of a spline is most often locally given by a low degree polynomial, even though the global structure is not a low degree polynomial. B-splines are polynomial splines that are defined in a way that resembles other basis function formulations [2]. Each single-variable function in ATLAS is approximated with uniform cubic B-spline basis functions, shown in Figure 2. B-splines can approximate any single-variable function, similar to using the Fourier basis. With uniform B-splines, each basis function is scaled so that the unit interval is **uniformly** partitioned, as in Figure 2.

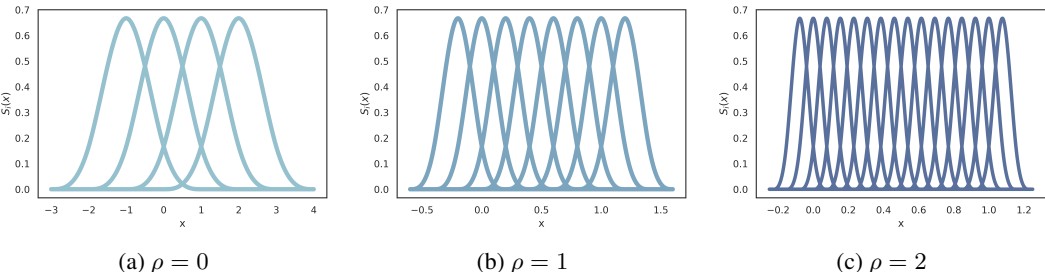

(a) $\rho = 0$          (b) $\rho = 1$          (c) $\rho = 2$

Figure 2: If uniformly spaced B-splines are used, then each basis function has the same shape. This makes it possible to use the same activation function by scaling and translating the inputs. This is also true for different densities of uniform cubic B-splines.

The activation function to implement B-splines is given by:

$$S(x) = \begin{cases} \frac{1}{6}x^3 & 0 \le x < 1 \\ \frac{1}{6}\left[-3(x-1)^3 + 3(x-1)^2 + 3(x-1) + 1\right] & 1 \le x < 2 \\ \frac{1}{6}\left[3(x-2)^3 - 6(x-2)^2 + 4\right] & 2 \le x < 3 \\ \frac{1}{6}(4-x)^3 & 3 \le x < 4 \\ 0 & otherwise \end{cases}$$

The choice was made to use uniform cubic B-splines due to their excellent performance and robustness to catastrophic forgetting, illustrated in Figure 3. Using uniform B-splines instead of arbitrary sub-interval partitions (also called knots in literature) makes optimisation easier. Optimising partitions is non-linear, but optimising only coefficient (also called control points) is linear and thus convex.

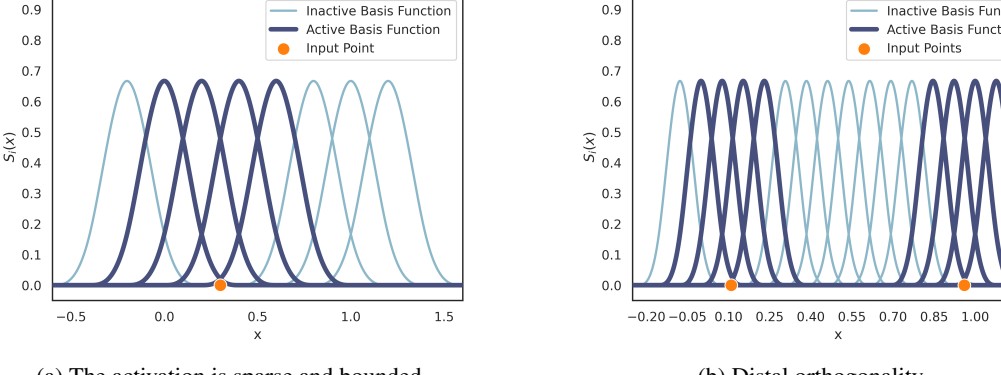

(a) The activation is sparse and bounded.          (b) Distal orthogonality.

Figure 3: Single-variable function

Each basis function is multiplied by a parameter and summed together. The total number of basis functions is typically fixed. Cubic B-splines are 3rd order polynomials, and thus require a minimum of $3 + 1 = 4$ control points or basis functions.

123 Instead of considering arbitrary densities of uniform cubic B-splines, we look at powers of two times
124 the minimum number of basis functions, called $\rho$-density B-spline functions.

125 **Definition 1** ($\rho$-density B-spline function)**.** A $\rho$-density B-spline function is a uniform cubic B-spline
126 function with $2^{\rho+2}$ basis functions:

$$f(x) = \sum_{i=1}^{2^{\rho+2}} \theta_i S_i(x) = \sum_{i=1}^{2^{\rho+2}} \theta_i S(w_i x + b_i) = \sum_{i=1}^{2^{\rho+2}} \theta_i S((2^{\rho+2} - 3)x + 4 - i)$$

127 Consider the problem of expanding a single-variable function approximator with more basis functions
128 to increase its expressive power. Using the Fourier basis makes it trivially easy by adding higher
129 frequency sines and cosines with coefficients initialised to zero. It is trickier to achieve something
130 similar with uniform cubic B-splines. There are algorithms for creating new splines from existing
131 splines with knot insertion, but the intermediate steps result in non-uniform knots and splines. A
132 simple and practical compromise that we propose is to use mixtures of different $\rho$-density B-spline
133 functions, as illustrated in Figure 2.

134 **Definition 2** (mixed-density B-spline function)**.** A mixed-density B-spline function is a single-
135 variable function approximator that is obtained by summing together different $\rho$-density B-spline
136 functions. Only the maximum $\rho$-density B-spline function has trainable parameters, the others are
137 constant. Mixed-density B-spline functions are of the form:

$$f(x) = \sum_{\rho=0}^{r} \sum_{i=1}^{2^{\rho+2}} \theta_{\rho,i} S_{\rho,i}(x)$$

138 Only the **maximum** $r = \rho$-density B-spline has trainable coefficients. All lower density $r > \rho$-
139 density B-spline have frozen and constant coefficients. The maximum $r = \rho$-density B-spline has
140 trainable coefficients with gradient updates that are orthogonal if the distance between two inputs is
141 large enough.

142 Similar to increasing the expressiveness of a Fourier basis function approximator by adding higher
143 frequency terms, one can add larger density cubic B-spline functions. Analytically, we can initialise
144 all the new scalar parameters $\theta_{r+1,i} = 0, \ \forall i \in \mathbf{N}$ such that:

$$f(x) = \sum_{\rho=0}^{r} \sum_{i=1}^{2^{\rho+2}} \theta_{\rho,i} S_{\rho,i}(x) = \sum_{\rho=0}^{r+1} \sum_{i=1}^{2^{\rho+2}} \theta_{\rho,i} S_{\rho,i}(x)$$

145 It is therefore possible to create a minimal model with $r = 0$ initialised at zero, and train the model
146 until convergence. Then one can create a new model with $r = 1$, by subsuming the previous model's
147 parameters, and train this more expressive model until convergence. This process of training and
148 expansion can be continued indefinitely, and is shown in Figure 8.

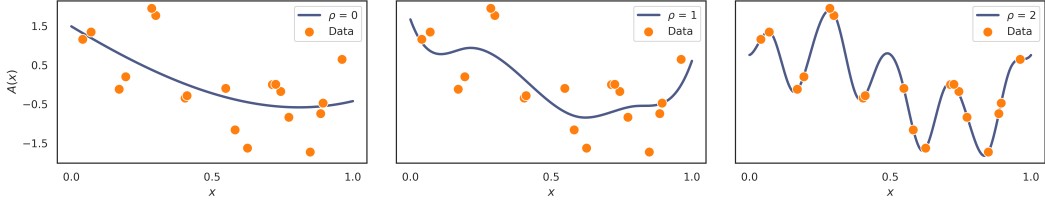

Figure 4: Doubling densities of basis functions before and after training.

## 6  ATLAS

150 ATLAS is named for carrying the burden of all it must remember, after the Titan god Atlas in Greek
151 mythology who was tasked with holding the weight of the world. ATLAS is also an acronym for
152 AddiTive exponentiaL Additive Splines.

**Definition 3** (ATLAS). ATLAS is a function approximator of $n$ variables, with mixed-density B-spline functions $f_j(x_j)$, $g_{i,j}(x_j)$, and $h_{i,j}(x_j)$ in the form:

$$A(\vec{\mathbf{x}}) := \sum_{j=1}^{n} f_j(x_j) + \sum_{k=1}^{M} \frac{1}{k^2} \exp\big(\Sigma_{j=1}^n g_{k,j}(x_j)\big) - \frac{1}{k^2} \exp\big(\Sigma_{j=1}^n h_{k,j}(x_j)\big)$$

ATLAS is equivalently given by the compact notation:

$$A(\vec{\mathbf{x}}) := F(\vec{\mathbf{x}}) + \sum_{k=1}^{M} \frac{1}{k^2} \exp(G_k(\vec{\mathbf{x}})) - \frac{1}{k^2} \exp(H_k(\vec{\mathbf{x}}))$$

The absolutely convergent series of scale factors $k^{-2}$ was chosen for numerical stability and to ensure the model is absolutely convergent. Another feature is that the series of scale factors also breaks the symmetry that would otherwise exist if all mixed-density B-spline functions were initialised to zero. Initialising all the parameters to be zero is a departure from the conventional approach of random initialisation. The number of exponential terms can be increased without changing the output of the model. We can choose to initialise $G_{M+1}(\vec{\mathbf{x}}) = 0$ and $H_{M+1}(\vec{\mathbf{x}}) = 0$, such that the model capacity can be increased at will.

ATLAS is a universal function approximator with some inherent memory retention. It possesses three properties atypical of most universal function approximators:

1. The activity within ATLAS is sparse – most neural units are zero and inactive.

2. The gradient vector with respect to trainable parameters is bounded regardless of the size and capacity of the model, so training is numerically stable for many possible training hyper-parameters.

3. Inputs that are sufficiently far from each other have orthogonal representations.

The proofs of the three properties follows from the single-variable case, the assumption of bounded single-variable functions and parameters, and the absolutely convergent $k^{-2}$ scale factors.

**Property 1** (Sparsity). *For any $\vec{\mathbf{x}} \in D(A) \subset R^n$ and bounded trainable parameters $\theta_i$ with index set $\Theta$, the gradient vector of trainable parameters (for ATLAS) is sparse:*

$$\left\| \vec{\boldsymbol{\nabla}}_{\vec{\theta}} A(\vec{\mathbf{x}}) \right\|_0 = \sum_{i \in \Theta} d_{Hamming} \left( \frac{\partial A}{\partial \theta_i}(\vec{\mathbf{x}}), 0 \right) \le 4n(2M+1)$$

*Remark.* For a fixed number of variables $n$, the model has a total of $n2^{r+2}(2M+1)$ trainable parameters. The gradient vector has a maximum of $4n(2M+1)$ non-zero entries, which is independent of $r$. Recall that only the maximum density ($\rho = r$) cubic B-spline function has trainable parameters. The fraction of trainable basis functions that are active is at most $2^{-r}$. Sparsity entails efficient implementation, and suggests possible memory retention and robustness to catastrophic forgetting.

**Property 2** (Gradient flow attenuation). *For any $\vec{\mathbf{x}} \in D(A) \subset R^n$ and bounded trainable parameters $\theta_i$ with index set $\Theta$: if all the mixed-density B-spline functions are bounded, then the gradient vector of trainable parameters for ATLAS is bounded:*

$$\left\| \vec{\boldsymbol{\nabla}}_{\vec{\theta}} A(\vec{\mathbf{x}}) \right\|_1 = \sum_{i \in \Theta} \left| \frac{\partial A}{\partial \theta_i}(\vec{\mathbf{x}}) \right| < U$$

*Remark.* For a fixed number of variables $n$, the model has a total of $n2^{r+2}(2M+1)$ trainable parameters. The factor of $k^{-2}$ inside the expression for ATLAS is necessary to ensure the sum is convergent in the limit of infinitely many exponential terms $M \to \infty$. Only the maximum density ($\rho = r$) cubic B-spline function has trainable parameters, so that the gradient vector is bounded in the limit of arbitrarily large densities $r \to \infty$. Smaller densities cannot be trainable, otherwise this property does not hold. The bounded gradient vector implies that ATLAS is numerically stable during training, regardless of its size or parameter count.

**Property 3** (Distal orthogonality)**.** *For any $\vec{\mathbf{x}}, \vec{\mathbf{y}} \in D(A) \subset R^n$ and bounded trainable parameters $\theta_i$ for an ATLAS model $A(\vec{\mathbf{x}})$:*

$$\min_{j=1,\ldots,n} \{|x_j - y_j|\} > 2^{-r} \implies \langle \vec{\boldsymbol{\nabla}}_{\vec{\theta}} A(\vec{\mathbf{x}}), \vec{\boldsymbol{\nabla}}_{\vec{\theta}} A(\vec{\mathbf{y}}) \rangle = 0$$

*Remark.* Two points that sufficiently differ in each input variable have orthogonal parameter gradients. Distal orthogonality means ATLAS is reasonably robust to catastrophic forgetting, without other regularisation and training techniques. However, memory retention can still potentially be improved when used in conjunction with other techniques.

ATLAS can be implemented with 1D convolution, reshaping, embedding, multiplication and dense layers. The same basis functions have to be computed for each input variable, hence 1D convolutions. By correctly scaling, shifting, and rounding inputs one can compute only the non-zero basis functions with embedding layers. The number of basis functions are chosen from powers of two for convenience, with the maximum density B-spline function having exactly $\lambda = 4 \times 2^r$ basis functions. Summing over all densities the total number of all basis functions in each input variable is at most $2\lambda$, because a geometric series was used. For every output dimension $p$, there are $2M$ exponentials. Each exponential has $n$ single variable functions, with at most $2\lambda$ cubic B-spline basis functions each. ATLAS models have time complexity $\mathcal{O}(pMn \log \lambda)$, and $\mathcal{O}(pMn\lambda)$ space complexity.

# 7   Methodology

The 1-,2- and 8-dimensional models were considered for evaluation, in combination with a chosen width for the update region in Task 2 from 0.1 to 0.9 in 0.1 increments. 30 trials were performed for each combination of model dimension and update region width. Mean Absolute Error (MAE) loss function, the Adam optimiser, and mini batch sizes of 100 are used throughout all experiments.

At the beginning of each trial (for a given dimension and update region width) a random learning rate was sampled uniformly between $10^{-6}$ and $0.01 + 10^{-6}$. A random noise level was sampled from an exponential distribution with scale parameter equal to one. The Task 1 target function is constructed from 1000 Euclidean radial basis functions (RBFs) with locations chosen uniformly over the entire input domain, with RBF scale parameters sampled independently from an exponential distribution (scale parameter equal to 10). The weights of each radial basis function are sampled from a normal distribution with mean zero and standard deviation equal to one. The Task 2 target function is exactly the same as the Task 1 target function – except for a square-like region with width equal to update region width. The location of the update region is chosen uniformly at random, and such that it is completely inside the domain of the model. The updated region masks the Task 1 target function and instead replaces the values inside it with another function that is sampled from the same distribution as the Task 1 target function, but independently from the Task 1 target function.

After the generation of the target functions 10000 data points are sampled for training, validation, and test sets for Task 1 and Task 2. To simulate the effect of learning unrelated tasks, the training data for Task 2 is only sampled from update region - with no training data outside of it being presented again, by contrast the validation and test sets for Task 2 were sampled over the entire input domain. Gaussian noise with standard deviation equal to the randomly chosen noise level is added to all training data. An ATLAS model ($M = 10$ positive and $M = 10$ negative exponential functions, maximum basis function density $r = 4$) with guaranteed distal orthogonality is trained and evaluated on Task 1 and Task 2. Then a modified ATLAS model ($M = 10$ positive and $M = 10$ negative exponential functions, maximum basis function density $r = 4$, trainable lower density basis functions) without guaranteed distal orthogonality is trained and evaluated on Task 1 and Task 2 using the same data sets as previously mentioned model. The final test errors for Task 2 are presented. A randomly selected trial of the 2-dimensional case is shown for visual inspection. The experiments presented in the main body of the paper were performed on Google Colab and the relevant code is provided.

# 8   Results

As shown in Figure 5 the effect of distal orthogonality is clear and crisp boundaries that limit the effect of Task 2 on the memory of Task 1. Without distal orthogonality there are more off-target effects that can be visualised.

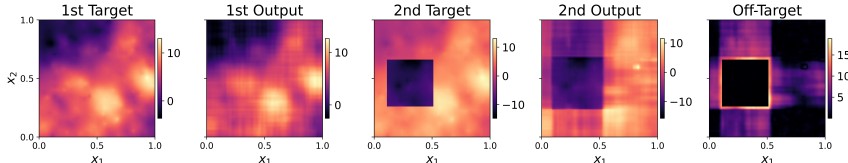

(a) Guaranteed distal orthogonality, Off-target effects deviate from Task 2 target.

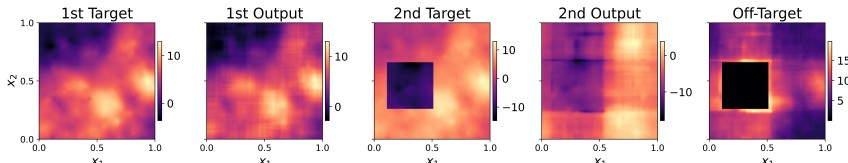

(b) No guaranteed distal orthogonality, Off-target effects deviate from Task 2 target.

Figure 5: A randomly chosen trial is presented for visual inspection.

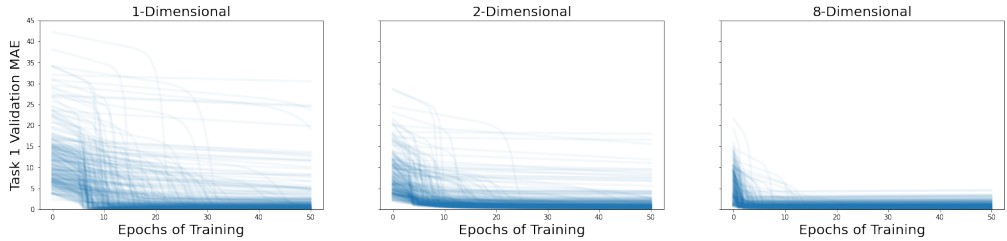

Figure 6: Distal orthogonality guaranteed: All validation MAE curves for Task 1.

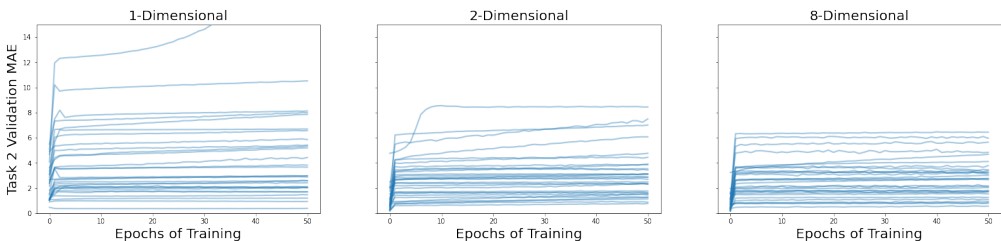

Figure 7: No distal orthogonality: Task 2 validation MAE with update region width $\delta = 0.1$.

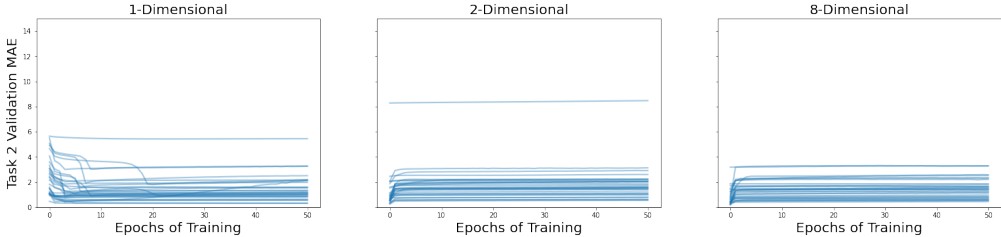

Figure 8: Distal orthogonality guaranteed: Task 2 validation MAE with update region width $\delta = 0.1$.

The effect of distal orthogonality on the averaged MAE for various trials for 1-,2- and 8-dimensional problems are presented as scatter plots of the averaged MAE over 30 trials for different update region widths as shown in Figure 9. The expected off-target error depends on the dimension of the problem and the width of the updated regions.

Analytical results to the expected off-target error require simplification, but a reasonable assumption in the absence of other evidence is that each input dimension has equal contribution on the unit hyper-cube. Assume for a fixed input dimension $n$ and some region of width $0 < \delta < 1$ where the

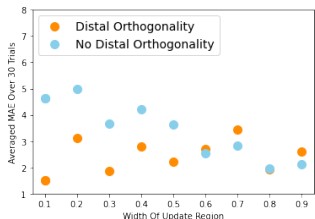

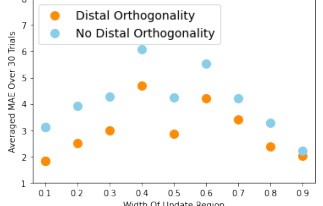

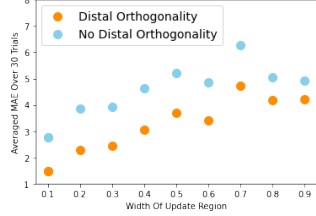

| (a) The 1-dimensional model. | (b) The 2-dimensional model. | (c) The 8-dimensional model. |

Figure 9: The effect of distal orthogonality on the final test error on task 2 for the 1-,2- and 8-dimensional input.

target function $Y$ is changed such that $|\Delta Y| = 1$ is one larger than it was originally. The expected off-target error depends on $k$ the number of input variables inside the updated region: $\varepsilon_k \approx \frac{n-k}{n}$. To correctly account for all permutations with the same magnitude of change:

$$p(\varepsilon_k) = \binom{n}{k} \delta^{n-k} (1 - \delta)^k$$

One can calculate expected change values:

$$\mathbb{E}[\varepsilon] = \sum_{k=0}^{n} \varepsilon_k p(\varepsilon_k) \approx \sum_{k=0}^{n} \left( \frac{n-k}{n} \right) \binom{n}{k} \delta^{n-k} (1 - \delta)^k = \delta$$

However if one assumes that the target function inside the updated region of width $\delta$ is correct, with probability $\delta^n$ of sampling from the entire input-domain, then the expected off-target error should be:

$$\text{Expected off-target error} \approx \delta - \delta^n$$

This seems consistent with some of the experimental results, but further investigation is needed.

# 9   Conclusion

The main contribution of the paper is theoretical and technical. A representation theorem is presented that outlines how to approximate multi-variable functions with single-variable functions (splines and exponential functions). ATLAS approximates all arbitrary single-variable functions with mixtures of B-spline functions. ATLAS is constructed in such a way that the gradient vector with respect to trainable parameters is bounded, regardless of how large an ATLAS model is. The activation of units in ATLAS is sparse, and allowed for an efficient implementation that only computes non-zero activation values with the aid of embedding layers. The gradient update vector with respect to trainable parameters is orthogonal for different inputs as long as the inputs are sufficiently different from each other.

For every output dimension $p$ in an ATLAS model, there are $2M$ exponentials. Each exponential has $n$ single variable functions, with at most $2\lambda$ cubic B-spline basis functions each. ATLAS models have time complexity $\mathcal{O}(pMn \log \lambda)$, and $\mathcal{O}(pMn\lambda)$ space complexity.

ATLAS was shown to exhibit some memory retention, without the assistance of other techniques. This is a good indication of the potential for combining it with other techniques and models for continual learning. The chosen experiments demonstrated the theoretically derived predictions and contrasted two models, incuding a variant of ATLAS without distal orthogonality guarantees.

As far as societal impacts are concerned: It is possible that ATLAS could allow for the creation of more powerful machine learning algorithms, that require less resources to train and deploy. Further testing is needed to make any concrete claim.

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
