# OpenReview forum: "Atlas: Universal Function Approximator For Memory Retention"
_NeurIPS.cc/2022/Conference — NeurIPS 2022 Submitted_

### Official Review · Reviewer_9ayK · 2022-06-29

**Rating:** 3
**Confidence:** 4
**Soundness:** 1 poor
**Presentation:** 1 poor
**Contribution:** 2 fair

**Summary:**

This work presents Atlas, a new universal function approximator based on B-splines. Theoretically, Atlas has nice properties, such as sparse gradients, bounded gradients, and distal orthogonality. Experimentally, Atlas is robust to catastrophic forgetting, indicating its potential to be applied in continual learning.

**Questions:**

In line 90, it is mentioned that "The minimum number of cubic B-spline basis functions is four." In line 140, it is written that "It is worth recalling that at most four basis functions are active for uniform cubic B-spline functions." Is there a contradiction here? Why four basis functions?

**Limitations:**

Some limitations are already listed in the Conclusion. Given that there is still space in the main paper, the authors are encouraged to finish some of them.

**Strengths And Weaknesses:**

As far as I know, the approach presented in this work is novel. This new function approximator has good theoretical properties with sound proofs. The experiments also show its potential in reducing catastrophic forgetting and continual learning.

However, this work can also be improved in many ways.

First, related works are not adequately cited. In continual learning, more strategies to mitigate catastrophic forgetting can be found in this [survey](https://arxiv.org/abs/1909.08383). Some methods with dynamic architectures should be cited and compared.
A short introduction of B-splines would be appreciated, given that B-splines form the basis of Atlas. As for memory retention, there are also some related works in deep learning, such as https://arxiv.org/abs/2008.13363 and http://arxiv.org/abs/2112.03257.

The notations used in the paper are not well-defined or explained. For example, in Definition 1, what are function $S_i(x)$ and $S(x)$? What is a uniform cubic B-spline function? Is $\theta_i$ a vector or a real value? There are similar issues in Definition 3 and Property 1-3. Providing proof sketches in the main paper for Theorem 2 and Property 1-3 would be much appreciated.

There is a lack of comparison of Atlas with traditional artificial neural networks for approximating the swiss-roll target function.
It will also be better to see the results of testing Altas in the classical continual learning setting on some benchmarks, such as MNIST.
Moreover, the experimental results are not well-explained. In Figure 6, why does the output difference have a cross shape?
I would also like to know the exact number of parameters used in the experiments for Atlas.

Overall, this work is interesting but more like a work in progress. And it can be improved significantly.

---

> ### Author Response · Authors · 2022-08-02
> **Answers and Revisions**
>
> Thanks for your feedback and suggestions for improvement. Many of the raised issues were considered in revising the paper.
>
> Vector quantities like $\vec{\mathbf{x}}$ are clearly indicated with a bar or arrow for legibility. Parameters, inputs, functions etc. without a bar or arrow are scalar quantities like $S(x)$. Some scalar quantities with indices are the scalar components of a vector like $x_{j}$ or scalar parameters in the model like $ \theta_{i} $
>
> We have updated the theory and introductory sections to include more details on how B-splines work, and why they were chosen drawing from the appendices and the cited works to better contextualise the theory in the main paper. B-splines were originally defined recursively, but uniform B-splines admit the possibility of being implemented with a piece-wise defined non-recursive activation function we denote $S(x)$. A Practical Review of Uniform B-Splines by  K Branson is a useful resource in understanding B-splines.

---

> ### Author Response · Authors · 2022-08-04
> **Specific Points of Interest**
>
> Dear Reviewer
>
> We would like to address specific points that had been raised.
>
> "First, related works are not adequately cited..." -- We have included a broader scope of related works in the relevant studies section, and read through the papers mentioned by you. In the context of the survey (https://arxiv.org/abs/1909.08383), our architecture can be described as using parameter isolation methods.
>
> "A short introduction of B-splines would be appreciated, given that B-splines form the basis of Atlas." -- Splines are piece-wise defined single-variable functions over some interval. Each sub-interval of a
> spline is most often locally given by a low degree polynomial, even though the global structure is not
> a low degree polynomial. B-splines are polynomial splines that are defined in a way that resembles
> other basis function formulations. Common descriptions in the literature regarding B-splines give recursive definitions and are obviously ill-suited for deep learning applications. However, if the sub-intervals are uniformly spaced, then each and every basis function has the same shape -- which means we can obtain any specific basis function by scaling and shifting the input. Just like one can get different sine and cosine basis functions/frequencies by scaling the input. This insight makes it possible to use a single activation function, with its inputs scaled and shifted correctly. This is taken from the cited work "A practical review of uniform b-splines" by K. Branson, and defined in our appendices. This activation function is denoted $S(x)$.
>
> "what are function $S_{i}(x)$ and $S(x)$" --The activation function $S(x)$ has a piece-wise definition (given in the appendices and cited works):
> $$0 \leq x < 1 \implies S(x) =  \frac{1}{6} x^{3}$$
> $$1 \leq x < 2 \implies S(x) =  \frac{1}{6} \left[-3(x-1)^{3} +3(x-1)^{2} +3(x-1) + 1 \right] $$
> $$2 \leq x < 3 \implies S(x) = \frac{1}{6} \left[3(x-2)^{3} -6(x-2)^{2} + 4 \right] $$
> $$3 \leq x < 4 \implies S(x) =  \frac{1}{6} ( 4-x ) ^{3} $$
> $$ x < 0 \text{ or } x \geq 4 \implies S(x) =  0 $$
>
> The function $S_{i}(x) := S(w_{i}x +b_{i}) $ is a basis function just like $sin(2 \pi n x)$ is a single-variable basis function. The specific scaling and shifting parameters/constants $w_{i},b_{i}$ dictate which basis function it is.
>
> "What is a uniform cubic B-spline function? " -- It is a piece-wise defined function over some interval, such that all sub-intervals/partitions are uniform or equally large.
>
> "Is $\theta_{i}$ a vector or a real value?" -- $\theta_{i}$ is a scalar parameter.
>
> "Providing proof sketches in the main paper..." -- We have added the necessary assumptions and additional visual proofs of the properties for the single-variable case in the main body of the paper (which is discussed and shown in cited works and the appendices). The multi-variable case follows from the single-variable case, but the complete proofs for models of $n$ input variables is roughly 10 pages of algebra detailed in the appendices.
>
> "There is a lack of comparison of Atlas with traditional artificial neural networks..." -- We have replaced the previous experiments with randomly sampled target functions composed of 1000 radial basis functions to replace the swiss-roll function. We also added a variant of our architecture that does not have guaranteed distal orthogonality to compare the effect of distal orthogonality. One model has guaranteed distal orthogonality, another does not have guaranteed distal orthogonality which is more typical of conventional ANNs. Some of the cited works already evaluated more conventional ANNs against splines and other piece-wise defined functions for memory retention.
>
> "Why four basis functions?" -- Cubic B-splines are locally defined with 3rd order polynomials, that require 3 + 1 parameters. Thus there are four parameters and four corresponding basis functions that are needed locally at any point inside the support interval.
>
> The minimum number of uniform cubic B-spline basis functions is four, but one could use a model with more than a billion basis functions in total, and even then at most four will be non-zero/active for any input value on the support interval. This is where the sparsity stems from.
>
> "Overall, this work is interesting but more like a work in progress. And it can be improved significantly." -- Thank you for your feedback which helped improve our work and presentation.

---

> > ### Comment · Reviewer_9ayK · 2022-08-05
> > **Thank you for the clarifications**
> >
> > Thank you for the clarifications.
> > I still have some concerns that I hope could be addressed.
> > Personally, I think the key factors of all continual learning algorithms are performance, computation efficiency, and memory efficiency. If memory and computation constraints are not considered, we could simply store all previous training samples and retrain a model again and again whenever receiving a new sample, in order to achieve a good performance.
> > Given that, I would be more convinced to see a comparison of Altas and traditional artificial neural networks in the classical continual learning setting on some benchmarks. Can it achieve comparable or better performance compared ANNs? What is the number of parameters used for Atlas? Is it more computation-efficient in terms of wall clock time?

---

> > > ### Author Response · Authors · 2022-08-06
> > > **Computational Efficiency**
> > >
> > > Dear Reviewer
> > >
> > > Thank you for your interest and questions.
> > >
> > > Computational complexity and practical utility were considerations in developing ATLAS.
> > >
> > > Suppose that one knew exactly what number of basis functions to use over every input variable, then one could delete a few lines of code and use simple uniform cubic B-spline functions, instead of a mixture of different uniform cubic B-spline functions. Suppose the exact number of uniform cubic B-spline functions to use is $\lambda$. Using embedding layers and convolution layers it would take $\mathcal{O}(1)$ time steps to evaluate a single-variable uniform cubic B-spline function and $\lambda$ basis functions that require $\mathcal{O}(\lambda)$ memory. This would be true for every single-variable function in the model. The sum of single variable functions $\sum_{j=1}^{n} f_{j}(x_{j})$ would take $\mathcal{O}(n)$ time and $\mathcal{O}(n \lambda)$ memory. There are $2M$ exponential activation functions (including the choice of none/zero), and the argument of each is a sum of single variable functions. The overall complexity would be $\mathcal{O}((1+2M)n = \mathcal{O}(Mn) $ in time and  $\mathcal{O}((1+2M) n \lambda  = \mathcal{O}(  M n \lambda) $  in space. If the target function has $p$ output dimensions then the overall complexity for vector-valued outputs is $\mathcal{O}(pMn)$ time and $\mathcal{O}( p Mn \lambda) $ in space.
> > >
> > > Models that are initialised with many basis functions might not generalise well, and require more data, augmentation, and regularisation techniques. See the 1990 NeurIPS/NIPS paper (https://proceedings.neurips.cc/paper/1990/hash/94f6d7e04a4d452035300f18b984988c-Abstract.html) for other B-spline models that are capable of learning and expanding to train on data in a way that can generalise very well. The memory retention of such large and sparse models is better than smaller models and can be improved with additional training techniques. In practice, we do not know exactly how many basis functions to use. Suppose one attempted hyper-parameter optimisation and architecture search. There are algorithms for increasing the number of basis functions at will, but after training one cannot easily go back to the previous model without losing information - this makes back-tracking more difficult in searching for near-optimal configurations. One could checkpoint and save models, but this adds a lot of error-prone coding complexity. A simpler solution is to encapsulate this functionality in a single model class using a mixture of uniform cubic B-spline functions. We restrict the number of basis functions to be powers of two such that $\lambda=4\times 2^{r}$, and the total number of basis functions (including lower densities) is $ \approx \lambda + \frac{1}{2} \lambda + \frac{1}{4} \lambda...\leq 2 \lambda$. The computational complexity for ATLAS is now $\mathcal{O}(pnM\log\lambda) = \mathcal{O}(pnMr)$ in time and $\mathcal{O}(pnM\lambda) =\mathcal{O}(pnM2^{r})$ in space.
> > >
> > > Keep in mind that the original construction from polynomials hides how rich the model's representational power is. Because we use arbitrary continuous single-variable functions, it means the model can learn fractional powers or negative powers which is far more expressive than multi-variate polynomials. We rarely needed more than $M=10$ positive and negative exponential functions -- even if the target function was a sum of 1000 randomly generated radial basis functions.
> > >
> > > We took the feedback from all reviewers and drafted more thorough experiments, including random radial basis functions to demonstrate some of the effects that emerge as the dimensionality increases that have already been typed, formatted and added to the paper. Preliminary results on MNIST digits and fashion data sets indicate that the larger models can memorise most of the training data in a single epoch of training. There is a large generalisation gap between testing and training accuracy for large models, and (depending on the size) the model retains information on the previous task's training set when training on MNIST digits after being trained on fashion MNIST images. We'll have the final results and confirmation in ~3-4 days. A single trial takes a few minutes to ~20 minutes to run on a laptop but doing the same experiment with different hyperparameters and models takes much longer and requires more computing power.

---

> > > > ### Comment · Reviewer_9ayK · 2022-08-07
> > > > **Thank you for your further clarification**
> > > >
> > > > I would be very interested in to see experimental results which compare the performance, the number of parameters, and the wall clock running time of ATLAS and the traditional methods in deep learning. Here are two papers that might be useful
> > > > - http://arxiv.org/abs/1810.12488
> > > > - http://arxiv.org/abs/1904.07734

---

> > > > > ### Author Response · Authors · 2022-08-09
> > > > > **Preliminary Feedback On Additional Experiments**
> > > > >
> > > > > Dear Reviewer
> > > > >
> > > > > We worked through the two papers you specifically asked for and identified two experiments that would be most insightful: Incremental Domain Learning, and Incremental class learning, from the paper "Re-evaluating Continual Learning Scenarios: A Categorization and Case for Strong Baselines." These benchmarks do not give the model information about which task is being performed.
> > > > >
> > > > > Incremental domain learning on the MNIST data that uses only two labels/dimensions for the output yielded a test accuracy of approximately 65 % (50% is essentially random) which is comparable to the L2 regularisation method (approximately  66%) mentioned in the relevant paper.
> > > > >
> > > > > Incremental class learning with ATLAS yielded a test accuracy of approximately 20% (10% is essentially random), which is comparable to L2 regularisation and other gradient optimisers without the use of regularisation techniques (approximately 20%).
> > > > >
> > > > > Another experiment we considered was training on the MNIST digits data, and then the MNIST fashion data set. What is curious is that ATLAS retained information of MNIST digits after being trained on MNIST fashion. With accuracies on the previous 10 class classification problem approximately 40% (depending on the model size), and accuracies on the new problem being approximately 99% (10% is uniformly random).
> > > > >
> > > > > Further investigation showed that the metric/distance related to distal orthogonality guarantees is not a norm-based metric but instead given by:
> > > > >
> > > > > $$ d_{r}(\vec{x},\vec{y}) = \frac{1}{n} \sum_{i=1}^{n} \min ( 1, 2^{r} |x_{i}-y_{i} |) $$
> > > > >
> > > > > Distal orthogonality occurs if  $  d_{r}(\vec{x},\vec{y}) \geq 1$, but for intermediate distances between zero and one it does affect training dynamics and memory retention.
> > > > >
> > > > > The distance between MNIST digits from different classes is smaller than the distance between MNIST digits and MNIST fashion images, in most cases. This is probably why memory retention results seem better between training on MNIST digits and MNIST fashion when compared with incremental learning benchmarks based on subsets of the unpermuted MNIST digits dataset.
> > > > >
> > > > > Finalising, typesetting and formatting the experiments on the MNIST digits and fashion data sets and adding them to the revised paper will take roughly 18 hours.

---

### Official Review · Reviewer_qV8i · 2022-07-06

**Rating:** 3
**Confidence:** 4
**Soundness:** 2 fair
**Presentation:** 3 good
**Contribution:** 2 fair

**Summary:**

This paper introduces Atlas, a NN architecture based on mixed-density B-spline functions. These function satisfy a universal approximation theorem (proved in the paper), and have several useful properties, such as sparse and bounded gradients. The goal of this paper is to use such networks for continual learning problems.

**Questions:**

- What is meant by robust, and how does Atlas achieve such robustness while other NN models do not? Is there a quantitative metric associated to this robustness?

- How does Atlas compare with baseline models and continual learning methods, e.g. in terms of new task performance, memory (since Atlas is an expanding model)? In general, as similar model expansion and weight fixing ideas have been implemented in other NNs, it is not clear how one may benefit from Atlas. Are there problems Atlas can solve, but other NNs can not solve?

- Can Atlas be applied to some of the benchmark datasets for CL, for example permuted MNIST is fairly common, but more challeneging benchmarks also exist.

- How is information shared across tasks in Atlas? When the model is expanded, does the model rely heavily on previously learned information, or simply learn new tasks from scratch in the expanded parts of the model? How does the expanded learning on subsequent tasks compare to learning that new tasks from scratch?


**Limitations:**

Yes, some limitations are briefly mentioned in the conclusions.

**Strengths And Weaknesses:**

Strengths:
- The examination of this class of functions in interesting, and the authors explain these models and their properties clearly

- Several interesting properties of these functions are clearly proved in the text

Weaknesses:
- ATLAS is discussed as having memory retention, but this seems to come mainly from training one B-spline at a time and fixing previous splines after training. As such the memory retention appears to come from fixing model parameters, rather than something intrinsic to the models. Similar ideas were used in other continual learning papers, e.g. continual learning via neural pruning [1903.04476]

- B-splines are discussed as being "robust", but what is meant by robust is not clearly defined. If this refers to memory retention, then as stated in the previous comment, this memory retention seems to come from parameter fixing during training. Why would one need B-splines in this case?

- To address continual learning, e.g. sequentially learning tasks as is discussed in experiments, the ability to learn new tasks seems to come from expanding the model. Is this correct? This can also be done with standard NN models, e.g. as in Progressive Neural Networks [1606.04671]

- The experiments are only examined on 2D problems, but there are several reasonably standard benchmarks for sequential task learning. Moreover, no comparison to baseline models is shown.  This would seem especially relevant, as the Atlas models do not seem to robustly fit several tasks.

- There is no discussion or examination of information sharing across tasks, e.g. ideas such as forward or backward utilization or transfer of information across tasks. This would be helpful in understanding how the model could robustly use previously learned information across tasks.

- the related work section does not discuss the breadth of work on continual learning, including work on models with similar ideas implemented (such as the two mentioned above)

---

> ### Author Response · Authors · 2022-08-02
> **Revisions and Answers**
>
> Thank you for your time and feedback it is sincerely appreciated. We have incorporated some of your points into the revised paper.
>
> It is not necessary to expand ATLAS models to accommodate new tasks. If the model is large enough at the point of creation, then it exhibits distal orthogonality. We decided to replace the previous set of experiments with a simpler demonstration with randomly generated target functions and no expansion of the model whatsoever.
>
> There are numerous possible metrics for robustness. The contextual meaning for robustness was meant to be "it exhibits some memory retention without additional ad hoc training and regularisation techniques, and regardless of the choice of hyperparameters"
>
> Information sharing among tasks isn't explicitly engineered into the outlined architecture, but it might be possible and further investigation is warranted.

---

> ### Author Response · Authors · 2022-08-04
> **Specific Points of Interest**
>
> Dear Reviewer
>
> We would like to comment on some of the raised points and questions.
>
> "...memory retention appears to come from fixing model parameters, rather than something intrinsic to the models. Similar ideas were used in other continual learning papers, e.g. continual learning via neural pruning..." -- This is partially correct. It isn't necessary to increase the size of the ATLAS model to learn new tasks, as long as the model is large enough with a large density of basis functions that partition the unit interval, then it will exhibit distal orthogonality. To be fair: The lower density basis functions must have constant and untrainable parameters, otherwise, distal orthogonality will not hold. We sought a simple and easy way to move between different densities of basis functions at will and eventually settled on a geometric series of powers of two. The lower density parameters for large models are initialised at zero and never change during training, so they do not affect models that are large at the point of creation.
>
> We replaced the previous experiments and decided not to alter the model size or structure during training to alleviate any doubts.
>
> "B-splines are discussed as being "robust", but what is meant by robust is not clearly defined..." -- Distal orthogonality limits the number of off-target effects introduced when training on a second task. This is easy to visualise in lower dimensions, in higher dimensions one would have to resort to some metric for measuring catastrophic forgetting. We specified the mean absolute loss function and the Task 2 target function such that the validation and test loss for Task 2 indicate the off-target side-effects of learning on a subset of the entire domain.
>
> The motivation for looking at two tasks is because if one can guarantee error bounds for going from one task to the next, then one can look at arbitrarily many tasks with repeated application of the same analysis.
>
> "The experiments are only examined on 2D problems" -- we have extended the experiments to higher dimensional models and randomly sampled target functions composed of 1000 radial basis functions, instead of the simple swiss-roll function. We have also randomised most of the hyperparameters used for training, which are randomly sampled for each and every trial.
>
> "Why would one need B-splines in this case?" -- Any piece-wise defined single-variable function approximator that does not share or reuse parameters over large regions of the input space would suffice. We chose cubic B-splines because we want a smooth model.
>
> "There is no discussion or examination of information sharing across tasks..." -- Information sharing across tasks might be too high-level an abstraction for making sense of the mechanisms that our architecture utilises. The main mechanism we were concerned with was parameter isolation and improved memory and creating models that do not scale exponentially with the input dimension. One can get perfect memory retention using grids and lookup tables that partition the input domain into fully independent little hypercubes partitioning each coordinate into $d$ pieces one would need $d^{n}$ cubes and values for each cube in such a lookup table. It isn't clear how to quantify and measure such information exchange since classical information theory is difficult to apply in the analysis of ANNs. Naftali Tishby has some interesting ideas on applying information theory to the flow of information in ANNs.
>
> "the related work section does not discuss the breadth of work on continual learning, including work on models with similar ideas implemented" -- We have included a broader scope of relevant works in the relevant studies section. The paper on Continual Learning via Neural Pruning was especially insightful. The idea of graceful forgetting is very similar to the behaviour exhibited by ATLAS in practice while learning a new task.
>
> "What is meant by robust, and how does Atlas achieve such robustness while other NN models do not? Is there a quantitative metric associated to this robustness?" The second task's target function encapsulates the ideal behaviour with no side effects or off-target changes, in conjunction with mean absolute error it forms a metric for catastrophic forgetting.
>
> "How does Atlas compare with baseline models and continual learning methods, e.g. in terms of new task performance, memory (since Atlas is an expanding model)?" -- ATLAS is not state-of-the-art in and of itself for continual learning problems. It is a solid foundation for combining with other techniques to mitigate catastrophic forgetting.
>
> "Are there problems Atlas can solve, but other NNs can not solve?" -- The newer and improved implementation of ATLAS does not evaluate inactive basis functions, which means it is exponentially faster than the previous version from May. That efficiency will be useful.

---

### Official Review · Reviewer_LHt4 · 2022-07-14

**Rating:** 5
**Confidence:** 4
**Soundness:** 3 good
**Presentation:** 3 good
**Contribution:** 3 good

**Summary:**

In this work, the authors propose a theoretical model that is generally well-suited for continual learning and exhibits less forgetting. In particular, they present a novel universal approximator theorem with multi-variable functions using single-variable functions and also exponential functions. They conduct a set of experiments to back their theoretical findings.

**Questions:**

*Authors should make a comparison with [1-3] and show how current theoretical results proposed in this manuscript compares and also comment on the generalization guarantee of the theorem.

* Line 27, the authors have stated that ANNs have issues such as vanishing and exploding gradients, whereas this is not true. This is not an issue with ANNs but learning approach which is backpropagation of error (BP or BPTT), recently people have investigated local learning approaches such as Local representation alignment (LRA) and Difference target propagation (DTP and DTP-sigma), where they show models can be trained from zero weight initialization. Finally, ANNs by design are not well suited for continual learning, but additional memory (Gradient episodic memory, mnemonics), regularization-based approaches such as elastic weight consolidation (EWC), and sparsity helps them handle these tasks. I would advise the authors to revise the intro and avoid any partially correct claims.
Authors have reported they use test splits as the validation set to evaluate model performance on the test set. The reason we have a validation set is to approximate a distribution that we might see in the future, therefore we try all permutations and combinations to extract the best model. A common practice is to test your model once on test splits. Constantly changing hyper-parameters by checking performance on the test set is invalid. Hence I would request authors redo experiments with separate splits. – Since using a 0.01 learning rate with Adam is an odd choice. Additionally, there is no hyperparameter optimization and results with K-trials. One should conduct results across trials and report average performance with standard error. Hence I am not convinced by the experiments

* A second ablation study with various noise levels is needed. How does model performance change when the standard deviation for Gaussian noise is varied from 0.1 to some other constant?

* For task two why are training sets sampled over a domain [0.45-0.55], what are these magic numbers, please provide reasoning on how should one derive such ranges?

* Task 1 is trained for 30 epochs, whereas Task 2 is trained for 6 epochs, why is this and how did you arrive to such conclusions, given there are no stopping criteria and using only 6 epochs is an odd choice.

* I liked the theoretical finding on the sparsity of atlas which states trainable parameters for the atlas is sparse, this would help in understanding why models such as Hard attention to task (HAT) and Sparse neural coding network (SNCN) work well on continual learning benchmarks without any memory component. What is the theoretical bound or upper bound for the total number of neurons required that will guarantee generalization and less forgetting? Can the authors comment on this?

* [1] https://proceedings.mlr.press/v130/doan21a.html
* [2] https://arxiv.org/abs/2008.02219
* [3] https://openreview.net/forum?id=hecuSLbL_vC


**Limitations:**

Highlighted above

**Strengths And Weaknesses:**

# Strengths
* Approach and theorems are novel.
* Paper is well-written with fixable errors
* Results are okay

# Weakness
* Ablation study is missing
* No baseline models to compare model performance
* Experiment section is missing few details

---

> ### Author Response · Authors · 2022-08-02
> **Revisions and Answers**
>
> Thank you for your thorough and useful review. The feedback helped identify problems in the paper. We have incorporated many of your suggestions into the paper.
>
> One of the primary goals of the experiments is to demonstrate that the model possesses distal orthogonality - as predicted from theoretical considerations. We have replaced the experiments with randomly chosen hyperparameters (learning rate, gaussian noise level etc.) and target functions created from 1000 randomly initialised radial basis functions, and contrasted the results with a model that does not possess distal orthogonality. The update region is now randomly placed within the domain of the function instead of a constant [0.45,0.55].
>
> Since May we have been able to create an implementation that does not evaluate basis functions that are zero, with the help of embedding layers. Its time complexity is exponentially better than the previous version. The new time complexity is $\mathcal{O} (\rho M n)$, and space complexity $\mathcal{O} (2^{\rho} M n)$.
>
> We suspect sparsity could help larger hard-attention models retain information. Regarding orthogonal gradient descent (OGD), one can describe the continual learning problem in terms of a distribution over the input space, target values, and time (or order of task). OGD attempts to orthogonalise gradient updates across time. Distal orthogonality in contrast means if input points are far enough from each other in space, then they are orthogonal. Roughly speaking if the data distribution shifts to different parts of the input domain, then one gets orthogonalised gradient updates across time without any additional computational effort.

---

### Meta-Review · Area_Chair_8c8t · 2022-08-27

**Recommendation:** Reject
**Confidence:** Certain

**Metareview:**

The submission proposes a novel type of neural network based on B-splines and exponential functions designed to reduce catastrophic forgetting, proves universal approximation results, and provides some experimental results. The reviewers find the submission interesting, but believe that the submission could be improved significantly in a number of respects, including better practices in use of the test dataset, comparison against baselines, and distinguishing between properties of the network class and properties of the learning method. Accordingly, I cannot recommend the present paper for acceptance.

**Award:**

No

---

### Decision · Program_Chairs · 2022-09-14

Reject